# NLRP3 Inflammasome Activation in Hemodialysis and Hypertensive Patients with Intact Kidney Function

**DOI:** 10.3390/toxins12110675

**Published:** 2020-10-26

**Authors:** Christof Ulrich, Susann Wildgrube, Roman Fiedler, Eric Seibert, Leonie Kneser, Sylvia Fick, Christoph Schäfer, Silke Markau, Bogusz Trojanowicz, Matthias Girndt

**Affiliations:** 1Department of Internal Medicine II, Martin Luther University Halle-Wittenberg, 06120 Halle (Saale), Germany; susannwildgrube@gmx.de (S.W.); roman.fiedler@uk-halle.de (R.F.); leonie@kneser.de (L.K.); sylvia.fick@uk-halle.de (S.F.); christoph.schaefer@uk-halle.de (C.S.); Silke.markau@uk-halle.de (S.M.); bogusz.trojanowicz@uk-halle.de (B.T.); matthias.girndt@uk-halle.de (M.G.); 2Nephrologische Kooperation Villingen-Schwenningen, 78052 Villingen-Schwenningen, Germany

**Keywords:** hemodialysis, hypertension, caspase-1, monocytes, inflammation

## Abstract

Hypertension is not only an integrative characteristic of hemodialysis (HD) patients but is also very common in the general population. There is evidence that the inflammatory cytokine IL-β, regulated by the NLRP3 inflammasome via caspase-1, contributes to the hypertensive setting. Therefore, we investigated in an observational pilot study whether IL-1β secretion and inflammatory cell death (pyroptosis) are different in HD and hypertensive patients with intact kidney function. Twenty HD patients were age-, gender-, and diabetes-mellitus-matched to patients with hypertension and intact kidney function. Caspase-1 activity and pyroptosis rates were measured by flow cytometry. IL-1β was determined by qPCR and the ELISA technique. The inflammatory status (CRP) did not differ between both groups; however, the body mass index, a classical cardiovascular risk factor, was significantly elevated in blood pressure (BP) patients. BP patients had a higher frequency of caspase-1-positive monocytes compared to HD (*p* < 0.001). IL1-β protein secretion was significantly enhanced in BP, but ex vivo stimulation of blood cells resulted in higher pyroptosis rates in HD compared to BP patients (*p* < 0.01). Therefore, HD and BP patients differ in the extent of the NLRP3 inflammasome activation. The consequences of overweight, present in BP patients, may contribute to the significantly higher inflammasomal induction level. Whether low pyroptotic rates are equivalent to a dysfunctional immune response or a high pyroptotic output corresponds to over-activation remains to be clarified.

## 1. Introduction

Arterial hypertension is an essential cardiovascular risk factor. Exacerbated by a sedentary lifestyle and overweight/obesity, high blood pressure is a leading cause of death [1,2]. Hemodialysis patients are multi-morbid, facing many cardiovascular-related handicaps, including hypertension and atherosclerosis [3]. Both diseases are interrelated; hypertension hemodynamically causes renal damage, and chronic kidney failure influences blood pressure by hormonal mechanisms. There is evidence that inflammatory cytokines like IL-1β and IL-6 are among the factors driving both diseases [4,5,6], thus initiating and sustaining a kind of chronic low-grade inflammation in blood vessels and kidneys. IL-1β belongs to the “early response” cytokines, which are released at very early stages of an immune response and are able to stimulate other cytokines.

The importance of IL-1β is further highlighted by the fact that it underlies tight control mechanisms. Induction and secretion of IL-1β is organized in large protein platforms, so-called inflammasomes. Monocytes, macrophages, and granulocytes activated by pattern recognition receptors are triggered to assemble the specific inflammasome components. The NLRP3 inflammasome consists of the sensor protein (nucleotide-binding domain leucine-rich repeat (NLRP3)), adapter proteins (associated speck-like proteins (ASC)) and caspase-1. ASC has two domains, which are involved in its self-association. The so-called ASC speck formation is a distinguished feature of inflammasome activation [7] and provides a platform for recruitment of caspase-1, which in turn cleaves pro-IL-1β, thus inducing either the secretion of mature IL-1β or pyroptosis (a form of inflammatory programmed cell death) [8].

Therefore, activated caspase-1 contributes to tissue injury or repair. The NLRP3 inflammasome has a broad spectrum of activators, and the canonical activation pathway needs two signals for activation: firstly, NF-kB-dependent transcription of the NLRP3 and pro-IL-1β genes, and secondly, a danger signal to start complex assembly, which results in caspase-1 activation [9]. Various metabolic products including reactive oxygen species, ATP, nigericin, cholesterol crystals, urate crystal, and lipotoxic ceramide are known to function as caspase-1 activators. As the NLRP3 inflammasome is activated in both hypertension and end stage renal disease (ESRD) [10,11], we speculated that the extent of NLRP3 inflammasome activation may be different in patients with apparent hypertension and hypertensive, multi-morbid hemodialysis (HD) patients. Therefore, we investigated the inflammasome activation in 20 ESRD patients age-, gender-, and diabetes-matched to patients with hypertension and intact kidney function.

## 2. Results

### 2.1. Baseline Characteristics

The baseline characteristics of both cohorts are listed in Table 1. Typical electrolyte disturbances, together with elevated kidney-specific urea and creatinine levels, were observed in HD patients. HD and blood pressure (BP) patients did not differ regarding most classical cardiovascular risk parameters, yet body mass index (BMI) was significantly increased in BP patients. HD patients were normal, while BP patients were typically overweight (HD: median BMI 24.5 [20.8–39.2], BP: 28.8 [18.7–32.7], *p* = 0.022).

Although all patients suffered from high blood pressure, the prescription of β-blockers and diuretics was higher in HD (Appendix A). Furthermore, most of the HD patients needed three or more drugs for correction of blood pressure (HD: 70% vs. BP: 15%, *p* = 0.004).

### 2.2. NLRP3 Inflammasome Activation in Monocytes of HD and BP Patients

Caspase-1 is the central enzyme of the NLRP3 complex. Detection of the active molecule in monocytes can be considered as an activation event. The development of inhibitors binding to the active center of this enzyme simplifies its detection. Baseline data of caspase-1 staining demonstrate that a higher percentage of CD14+caspase-1+-positive monocytes are present in BP compared to HD patients (Figure 1a,b). As monocytes comprise a group of heterogeneous subsets that are phenotypically and functionally different, we analyzed classical (CD14++CD16-, Mo1), intermediate (CD14++CD16+, Mo2), and non-classical monocytes (CD14(+)CD16++, Mo3). Caspase-1 positivity was detected in all subsets, and there was a trend for higher frequencies for caspase-1-positive cells among the subsets of BP patients; however, only in Mo3 did the difference reach a significant level (Appendix A).

In the line with protein data, we found higher mRNA transcription levels of caspase-1 in PBMCs (peripheral blood mononuclear cells) of BP patients at baseline (HD: 0.73 ± 0.5 vs. BP: 2.2 ± 3.4, *p* < 0.05, Figure 1c). In addition, as caspase-4 and -8 may have an impact on the NLRP3 inflammasome, we measured the transcripts of both genes (caspase-4 expression; HD: 1.3 ± 1.4 vs. BP: 1.4 ± 1.1; caspase-8: HD: 0.7 ± 0.3 vs. BP: 0.8 ± 0.4, *p* = 0.459).

To validate the NLRP3 activation profile measured by caspase-1 analysis, we used the time of flight evaluation (TOFIE) assay, which measures ASC aggregates—the measurement of these so-called specks represents another possibility to analyze the induction of the NLRP3 inflammasome. The more pronounced ASC speck formation of BP patients is a further indication that BP patients have a higher NLRP3 inflammasome activation in comparison to HD patients under basal conditions (Figure 1d).

### 2.3. NLRP3 Inflammasome Activation: The Pyroptotic Route in HD Patients

Pyroptosis, a form of programmed cell death, is linked to caspase-1 activation. Pyroptotic monocytes can be analyzed by a combined cell staining with 7-AAD and caspase-1 (Figure 2a). This more differentiated analysis confirms on the one hand the higher frequency of CD14+caspase-1 monocytes in BP patients; on the other hand, however, we definitely see increased cell death rates in HD patients. Therefore, a significant number of caspase-1 activities move towards pyroptosis in HD patients (Figure 2b). In order to classify these results, we recruited three apparently healthy persons and age-matched them to HD and BP patients. The pyroptosis rate was 3.5 ± 1.7% for controls, 6.3 ± 3.6% for HD patients, and 0.6 ± 0.5 for BP patients (*p* = 0.08).

### 2.4. IL-1β Expression: Higher Levels of IL-1β in BP Patients

The main effector agonist of the NLRP3 inflammasome is the proinflammatory cytokine IL-1β. Therefore, we analyzed this cytokine in relation to protein and mRNA level. The higher IL-1β amounts measured in unstimulated samples of BP patients are reflective of the subtle differences determined by our caspase-1 and ASC assay (Figure 3a). Regarding the mRNA profile, there at least appears to be a trend for higher IL-1β mRNA levels in BP (Figure 3b). We also analyzed IL-6 transcripts. IL-6 mRNA levels are significantly elevated in BP in comparison to HD patients (HD: 0.7 ± 0.5 vs. 1.2 ± 0.9, *p* < 0.05). This result may be interpreted as an indication that—in spite of similar CRP levels in both cohorts—our overweight BP patients are fighting with a somewhat higher inflammatory activation compared to age-matched hemodialysis patients.

### 2.5. Full Activation of the NLRP3 Inflammasome: Pyroptosis in HD Patients

To shed some light on the NLRP3-specific stimulation process, we activated PBMCs of both groups with a mixture of LPS/nigericin mimicking both priming—the first step signal—and second step signaling to activate the NLRP3 complex. The classical features of such activation comprise caspase-1 activation, IL-1β secretion, and potential cell death. Regarding caspase-1 activation the flow cytometric analysis provided evidence that the cells of both groups responded to LPS/nigericin stimulation (HD—unstimulated (uHD): MFI 18.7 ± 4 vs. stimulated (sHD): 141.5 ± 60.4; uBP: 17.9 ± 5.0 vs sBP: 107.7 ± 36.0), but the caspase-1 expression was significantly higher in HD compared to BP patients (0.036). This, however, did not result in higher IL-β levels in the supernatants of PBMCs of both groups (sHD: 5409 ± 3166 pg/mL vs. sBP: 5507 ± 4620 pg/mL, *p* = 0.985). Therefore, we analyzed the rate of pyroptosis in both cohorts. As exemplified in the dot blots, caspase-1 activation in HD and BP patients is different (Figure 4a). Although 100% of monocytes of both cohorts stained positive for caspase-1 higher pyroptotic cell death rates were measured in hemodialysis in comparison to BP patients (Figure 4b). As caspase-4 and caspase-8 may contribute to pyroptosis, we again measured the transcripts of both genes. Under stimulatory conditions the expression of both genes, however, was not different between both groups (caspase 4 expression; stimulated HD: 2.8 ± 2.7 vs BP: 2.7 ± 2.2, 0.670; caspase-8: HD: 0.8 ± 0.4 vs. BP: 1.0 ± 0.6, *p* = 0.176).

We thought that the classification of different pyroptosis rates may be clarified by analysis of samples of healthy subjects. Data of age-matched controls show pyroptosis rates of 44.7 ± 7.8% (HD: 61.8 ± 22.2%; BP: 25.2 ± 32.0%), thus indicating that dual, sequential NLRP3 activation most probably shifts monocytes to pyroptosis, generating the condition for resolution of inflammation.

Activated caspase-1 has not only the potential to cleave pro-IL-1β but also gasdermin D. This is an important biological mechanism as the cleaved 30 kD fragment of gasdermin D molecules forms pores, a process that not only can mediate excretion of IL-1β but also lytic cell death (pyroptosis). Measuring both forms of gasdermin by capillary electrophoresis, we found on the one hand a significantly decreased level of intact gasdermin D (HD: 0.23 ± 0.17 vs BP: 0.50 ± 0.36; *p* = 0.036; Appendix A) and on the other hand an increased level of the cleaved form in HD compared to BP patients (HD: 0.50 ± 0.41 vs. BP: 0.24 ± 0.16; *p* = 0.065; Appendix A).

### 2.6. Intracellular ATP Measurement

The NLRP3 inflammasome is assembled as a large multiprotein complex—a process that consumes ATP. The intracellular ATP content (nmol/µL) in PBMCs of HD and BP patients ex vivo was not different at the basal level (HD: 0.024 ± 0.022 vs. BP: 0.022 ± 0.026, *p* = 0.396). Stimulation of PBMCs of both cohorts with LPS and nigericin, however, doubled the ATP content in HD patients, while ATP levels of BP patients remained at the basal stage (HD: 0.041 ± 0.040 vs. BP: 0.019 ± 0.019, *p* = 0.001).

### 2.7. Association between BMI and Caspase-1 Activity

Although correlation analysis in small cohorts should not be overemphasized, the corresponding analysis appears to make sense as the relationship of BMI and caspase-1 activity in both groups shows contrary tendencies. On the one hand there is a negative correlation of both parameters in HD patients (*r* = −0.352); on the other hand BMI and caspase-1 activity seem positively associated (*r* = 0.237) in BP patients.

## 3. Discussion

In this study we aimed to observe differences in the NLRP3 activation profile of two high cardiovascular risk groups: hemodialysis patients and hypertensive patients with intact kidney function. As it is well-established that not only kidney failure but also hypertension triggers the NLRP3 inflammasome induction [11,12,13], we matched HD and BP patients taking age, gender, and diabetes mellitus into account. Furthermore, we took care that the inflammation marker CRP was in the same range among matched partners. In spite of careful matching, the evaluation of the drug regime in both cohorts disclosed the different extent of cardiovascular stress in both groups. On average, three or more drugs were necessary to control blood pressure in HD patients. This, of course, can have an impact on immuno-active cells. As demonstrated by Shaw and colleagues, beta-blockers modulate the lymphocytic immune response by down-regulating the expression of CD107 and HLA-DR [14]. The impact of these drugs on monocytes, specifically on NLRP3 inflammasome activation, is unknown, and we do not know to what extent the higher NLRP3 activation in BP patients results from the different drug regime. Another parameter that may massively influence immune responses is related to the different weights of both cohorts.

As a matter of fact, HD and BP patients differed with regard to body mass index and kidney function. According to the “simplified” Framingham risk profiling, high BMI belongs to the group of cardiovascular risk factors predicting a worse outcome [15]. In recent decades it became clear that overweight/obesity is associated with inflammation—white fat produces many detrimental factors, including inflammatory cytokines [16]. Furthermore, it is known that monocytes are part of inflamed fat tissue. Rogacev and colleagues impressively demonstrated in their “I Like HOMe-study” the relationship of high numbers of CD16+ monocytes in obese patients with signs of subclinical atherosclerosis [17]. We now extend this observation in demonstrating that overweight/obesity is also associated with higher frequencies of CD16+caspase-1+ non-classical monocytes. The putative mechanism by which hypertension in combination with overweight drives caspase-1 expression in CD16+ monocytes has to be illuminated.

Hypertension is a life-threatening disease. However, the combination of overweight/obesity and hypertension may be even worse [18], strongly stimulating NLRP3 signaling. The chronic nature of both hypertension and kidney failure represents a constant challenge for the body, not only as far as immune-active cells are concerned. Therefore, overweight/obesity means an additional burden with broad effects on cytokine, hormone expression, lipid storage, and on the adipose-resident immune cell populations [19]. As well as involvement of caspase-1, caspase-8 and caspase-4 may also contribute to NLRP3 activation and pyroptosis [20,21,22], and we accordingly measured caspase-8 and caspase-4 transcripts in our cohorts. Caspase-8, for example, interacts with core components of the NLRP3 inflammasome and is required for caspase-1 activation [20], and caspase-4 is able to induce pyroptosis via gasdermin D cleavage, but, because neither caspase-4 nor caspase-8 transcripts were different in our cohorts, both proteases probably do not contribute to elevated pyroptosis measured in HD patients. One should, however, also keep in mind that there are still other mechanisms regulating pyroptosis. Rühl et al. impressively demonstrated that the endosomal sorting complex required for transport III (ESCRTIII)-dependent membrane system can delay or prevent pyroptosis [23]. Therefore, it is quite feasible to hypothesize that in PBMCs of hypertensive patients, the ESCRTIII system could be more active in comparison to that of HD patients. This, however, needs further investigations.

The classical feature of the NLRP3 inflammasome is to sense for microbial invaders or danger molecules, followed by an inflammatory response. Likewise, pyroptosis is an important feature of the NLRP3 inflammasome activation, for the lytic form of cell death eliminates infected immune cells while simultaneously destroying surviving bacteria by presenting them to phagocytes [24]. At first sight it seems plausible that one should expect elevated pyroptosis rates in the basal state in PBMCs of hemodialysis and hypertensive patients with intact kidney function. Under stimulatory conditions in vitro, there is the possibility for a dysfunctional, low pyroptotic, or an overshooting high pyroptotic response. Induction of the NLRP3 inflammasome with specific stimuli resulted in “low” (BP), “medium” (Co), and “high” pyroptosis rates (HD) in our patients, but it is quite obvious that a larger study is necessary to address this issue, as the number of age-matched controls included in our study was not sufficient to verify this hypothesis.

Hypertension is a leading cause of end stage renal disease [25]. Therefore, it is possible that the group of BP patients included in our study may develop renal disease in the future, and these patients may already carry undiagnosed signs of kidney disease at a very early stage. On that condition, low pyroptosis rates in BP patients could also be interpreted as adequate “low” NLRP3 inflammasome induction. From a theoretical point of view, a high pyroptosis rate guarantees total elimination of virus- and/or bacteria-infected cells, although over-activated pyroptosis can result in a massive inflammatory response. Thus, it is not clear if the “lower pyroptosis rate” is beneficial or undesirable. In the context of the very small control cohort, the low pyroptosis rate in BP patients may be too low, but of course the correct interpretation of these results deserves further clarification.

Regarding the question of what putative mechanism propels the NLRP3 inflammasome machinery in our cohorts, we speculated that intracellular ATP may play a role. There are several lines of evidence in the literature that ATP is involved in the regulation of inflammasome activation and programmed cell death in particular [26,27]. Indeed, ATP plays a role in cell-stress-induced NLRP3 inflammasome activation, and it is proven that extracellular ATP is a common inflammasome activating event [28]. The source of extracellular ATP for IL-1β release in vivo, however, is still questioned. It is suggested that dying or damaged cells release ATP to the extracellular space [29]. Unfortunately, we did not collect the supernatants in our in vitro incubation experiments to see if there are also different extracellular ATP levels in both cohorts. Recently, Nomura et al. demonstrated in bone marrow derived macrophages and THP-1 cells that the intracellular decrease in ATP concentration is linked to NLRP3 inflammasome activation [26]. This appears to contradict our findings, for in our model the stimulation of LPS/nigericin in PBMCs was followed by a significant rise in intracellular ATP. However, the experiments of Nomura et al. demonstrated the drop of ATP as early as 15 min after nigericin induction. The stimulation phase with nigericin in our model was terminated after 15 min exactly. Therefore, it is not clear at the moment whether the short stimulation time with the ionophore nigericin gives rise to ATP production prior to ATP depletion. Duncan and colleagues proved that formation of the NLRP3 complex consumes ATP; therefore, energy has to be provided by the cells [30]. Regarding energy supply and consumption, the focus is directed to inflammasomal activation events. Recently Lin et al. raised an interesting issue assigning the IL-1β receptor-associated kinase (IRAK-1) a role during NLRP3 activation [31]. They proved that simultaneous stimulation of toll-like receptors (e.g., microbes, LPS) and nucleotide-binding oligomerization domain-like receptors (e.g., microbial toxins, nigericin) led to rapid pro-caspase-1 cleavage without the need for transcription of pro-caspase-1 and NLRP3. The mechanism bypassing this classical NLRP3 priming appeared to be coordinated by IRAK-1. It contains an invariant lysine in its kinase domain, which is essential for ATP binding and catalytic function [32]. This pro-death pathway induced by dual stimulation appears to make sense, since it immediately eliminates the niche for survival and replication of pathogens. Although using a sequential stimulation model, our data lend support to the view that the “pyroptotic route” in inflammasome activation is activated in HD patients and impaired in overweight BP patients. However, this observation deserves greater clarification.

This leads us to the main drawbacks of our study, which was predominantly limited by low patient numbers. Furthermore, our study does not provide a clear answer as to which factors are responsible for the activation of the NLRP3 inflammasome in HD and BP patients. Nevertheless, planned as a pilot study, our data give insight in the different mode of NLRP3 activation in hypertensive, overweight, and hemodialysis patients.

## 4. Conclusions

The NLRP3 inflammasome is activated in obese patients with hypertension to a higher extent compared to hypertensive dialysis patients. Whether low pyroptosis rates in PBMCs of BP patients are an indication of an insufficient shutdown of the inflammatory loop or whether high pyroptosis rates of HD patients are part of an overshooting inflammatory response remains to be clarified.

## 5. Materials and Methods

### 5.1. Study Population

The observational study enrolled 20 end stage renal failure (HD) patients who were on maintenance hemodialysis for >6 months and 20 hypertensive patients (BP) with intact kidney function. HD patients were recruited from the outpatient unit of the Department of Internal Medicine II; patients with intact kidney function came from the nephrological and cardiological units of the hospital of Martin Luther University Halle-Wittenberg. Patients were >18 years of age, and patients with active malignancy, active infections (CRP > 50 mg/L), systemic autoimmune disorders (systemic lupus erythematosus, granulomatous polyangiitis), and neurologic disorders were excluded. Furthermore, HD patients suffering from gout—a well-known activator of the NLRP3 inflammasome—were also not included in the study. To prevent bias in the comparison of both groups, patients with intact kidney function were age-, gender-, and diabetes-matched to HD patients.

The causes of renal disease were diabetic nephropathy (*n* = 4), vascular nephropathy (n = 5), glomerulonephritis (*n* = 3), polycystic kidney disease (*n* = 3), interstitial nephritis (*n* = 3), nephrosclerosis (*n* = 1), and other/unknown (*n* = 1). For the classification of the disease-specific NLRP3 activation pattern, we recruited apparently healthy subjects (2 m/L f; 50.7 ± 11.0 years, 24.7 ± 2.3 BMI [kg/m^2^]).

All subjects enrolled were interviewed to determine cardiovascular diseases with a standardized questionnaire. In addition, patient charts were reviewed for cardiovascular events and drug regimes.

The study was conducted according to the Declaration of Helsinki. Written informed consent was obtained from all study subjects, and the study protocol was approved by the local ethics committee (No.:2017-27, Date: 10 April 2017, ethic committee of the Martin Luther University).

### 5.2. Clinical Procedures

Blood was taken under standardized conditions; in the case of HD patients, blood samples were drawn before the start of the dialysis. Creatinine, urea, hemoglobin, calcium, sodium, potassium, and CRP were measured in a certified clinical laboratory using routine methods. Patients with self-reported diabetes mellitus (DM) and with current use of antidiabetic drugs were categorized as diabetic.

### 5.3. PBMC Isolation

PBMCs were isolated from blood samples anti-coagulated with EDTA by ficol gradient centrifugation (GE Healthcare, Solingen, Germany). The vitality of PBMCs as tested by 7-AAD (Thermo Fisher Scientific, Darmstadt, Germany) staining was 99.4% ± 0.4 in HD and 99.8% ± 0.3 in BP patients. Contamination of PBMCs with granulocytes was 0.8% ± 0.9 in HD and 0.6% ± 1.3 in BP patients.

### 5.4. NLRP3 Inflammasome Stimulation Model

Cells (0.25–5 Mio, assay dependent), suspended in RPMI/2% FCS/1% glutamine (Sigma-Aldrich, Steinheim; Merck-Millipore, Darmstadt; PAA, Paching, Germany), were incubated at 37 °C, 5% CO_2_ atmosphere for 4 h. Lipopolysaccharides (LPS, 1 µg/mL, 0111:B4, Sigma-Aldrich,, Steinheim, Germany) and the microbial toxin nigericin (5 µg/mL, Sigma-Aldrich) were applied as stimuli (signal 1 and 2 for specific NLRP3 activation). In contrast to LPS, nigericin (acting as K^+^ ionophore) was applied for the last 15 min of the incubation period. Negative controls were run without stimulus.

### 5.5. Antibodies for Flow Cytometry

The following antibodies were used: anti-CD16 APC (clone 3G8, BD Biosciences, Heidelberg, Germany), -CD14PeCy7 (clone 61D3, Thermo Fisher Scientific), -CD15eF450 (clone HI98, Thermo Fisher Scientific), -CD3 Vioblue (clone BW264/56, Miltenyi Biotec, Bergisch-Gladbach, Germany), and 7-AAD (Thermo Fisher Scientific). Samples were analyzed using a MACS Quant analyzer (Miltenyi Biotec, Bergisch-Gladbach, Germany) with MACS Quantify software. Gates were set according to fluorescence minus one (FMO) controls.

### 5.6. Caspase-1-Assay

Caspase-1 was flow-cytometrically detected by a FAM-FLICA^®^ Caspase Assay using the FAM-YVAD-FLICA inhibitor probe as described by the manufacturer (BioRad, Feldkirchen, Germany). The fluorescent inhibitor binds to activated caspase-1. One hour before ending the regular incubation period, cells (0.25 Mio) were pelleted and re-suspended in sterile PBS/0.5% HSA containing the Casp-1 inhibitor probe. The incubation was continued for 1 h. Samples without caspase-1 inhibitor were used as negative controls. Cells were counterstained with anti-monocyte and lymphocyte-specific antibodies. For detection of pyroptosis, 7-AAD (Thermo Fisher Scientific) staining in combination with caspase-1 positivity was applied (AAD+Casp-1+).

### 5.7. Time of Flight Evaluation of ASC Specks by Flow Cytometry

ASC protein expression was investigated in isolated PBMCs (0.5 Mio). After surface staining of cells (anti-CD14 and -CD16), cells were permeabilized by saponin treatment (Sigma-Aldrich) prior to intracellular staining. ASC specificity in PBMCs was examined by staining with a specific anti-ASC clone (clone B3, AlexaFluor488 conjugated, Santa Cruz Biotechnology, Heidelberg, Germany). Time of flight evaluation (TOFIE) was performed according to Sester et al. [7] using the FSC–height vs. FSC–area blots.

### 5.8. Determination of Intracellular ATP

Cells were cultured as described in Section 2.4. Cell pellets were preserved in liquid nitrogen. ATP determination was performed by using the fluorogenic ATP Assay Kit (Abcam, Cambridge, UK). The samples were measured at 535/587 (excitation/emission) nm on an Infinite M200 Pro Analyzer (Tecan, Crailsheim, Germany).

### 5.9. RNA/cDNA/qPCR

RNA was isolated using the ZR RNA MiniPrep™ Kit (ZymoResearch, Freiburg, Germany). The RNA concentration and quality (260/280 ratio: HD: 2.0 ± 0.3; BP: 2.1 ± 0.1) was tested by Nanodrop technique (PEQLAB Biotechnologie GmbH, Erlangen, Germany). Equal amounts of RNA (100 ng) were reverse transcribed using a FastGene Scriptase Basic cDNA Kit (Nippon, Düren, Germany).

NLRP3 (Hs00918082_m1), caspase-1 (Hs00354836_m1), caspase-4 (Hs01031951_m1), caspase-8 (Hs01018151_m1), IL-1β (Hs00174097_m1), IL-6 (Hs00985639_m1), Pycard (Hs01547324_g1), and the expression of house-keeping genes ACTB (Hs01060665_m1) and RPL37a mRNA were analyzed using TaqMan probes (Thermo Fisher Scientific) using qPCRBIO Probe Mix High-ROX (Nippon, Düren, Germany). The samples were processed in duplicate on a StepOnePlus Cycler (Thermo Fisher Scientific). Data were normalized to house-keeping genes, related to healthy control donor RNA, and expressed as x-fold difference (2^^−ddCt^ method).

### 5.10. Simple Western Analysis

Cell lysates (RIPA buffer containing sodium-orthovanadate, PMSF, and protease inhibitor cocktail, Santa Cruz Biotechnology) were prepared from 5 × 106 PBMCs. Protein concentration was determined by a Protein Assay Kit (Bio-Rad Laboratories, Feldkirchen, Germany). Lysates (4.8 µg/6µL) were separated by capillary electrophoresis (Protein Simple, Wiesbaden, Germany) on a Sally Sue analyzer (Protein Simple). For Simple Western analysis (Protein Simple) the Separation Matrix Kit 2 (12–230 kD) was used. The kit includes 10 × sample buffer, stacking matrix 2, a separation matrix as well as upper and lower running buffers. Capillary electrophoresis was run on the corresponding size capillaries for Sally Sue (Protein Simple) analysis. For fluorescent labelling of samples, the standard pack 1 (Protein Simple) was used. Detection of specific proteins was performed by staining with anti-gasdermin D and anti-cleaved gasdermin D (Cell Signaling, Frankfurt, Germany) and anti-β-actin (Sigma-Aldrich) antibodies followed by staining with the anti-rabbit or anti-mouse detection module (Protein Simple). Data were analyzed by using the Compass software (Protein Simple Wiesbaden, Germany).

### 5.11. Statistics

Results are expressed as mean ± SD unless otherwise indicated. Categorical variables were statistically analyzed by Fisher’s exact test. All continuous variables were controlled for normal distribution using the D’Agostino–Pearson omnibus test. Continuous data were compared by the paired Student’s *t*-Test, Wilcoxon Test, or by one-way ANOVA followed by Sidak’s multiple comparisons or the Friedman test as appropriate. The relationship between body mass index (BMI) and caspase-1 activity was examined by Spearman correlation. All calculations were carried out using SPSS 21.0 (SPSS Inc., Chicago, IL, USA) or GraphPad Prism 6.0 Statistics software (GraphPad Software Inc., La Jolla, CA, USA). The level of significance was set at *p* < 0.05.

## Figures and Tables

**Figure 1 toxins-12-00675-f001:**
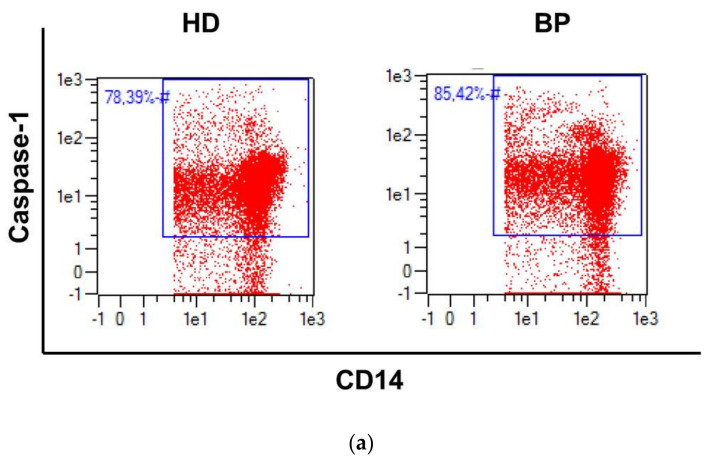
Activation status of NLRP3 components. (**a**) Representative dot blot of CD14+ monocytes staining positive for caspase-1 in hemodialysis (HD) patients and patients with high blood pressure (BP); (**b**) Frequency of CD14/caspase-1 double-positive monocytes in HD and BP patients; (**c**) caspase-1 mRNA expression (RQ, x-fold) in PBMCs of HD and BP patients; (**d**) aggregate analysis (TOFIE) of the apoptosis related speck protein containing a card domain (ASC) in PBMCs of HD and BP patients. The results are presented as box blots comprising median, 25th, and 75th percentiles. Statistical differences were analyzed by paired *t*-test (a) or the Wilcoxon test (b, c). * *p* < 0.05, *** *p* < 0.001.

**Figure 2 toxins-12-00675-f002:**
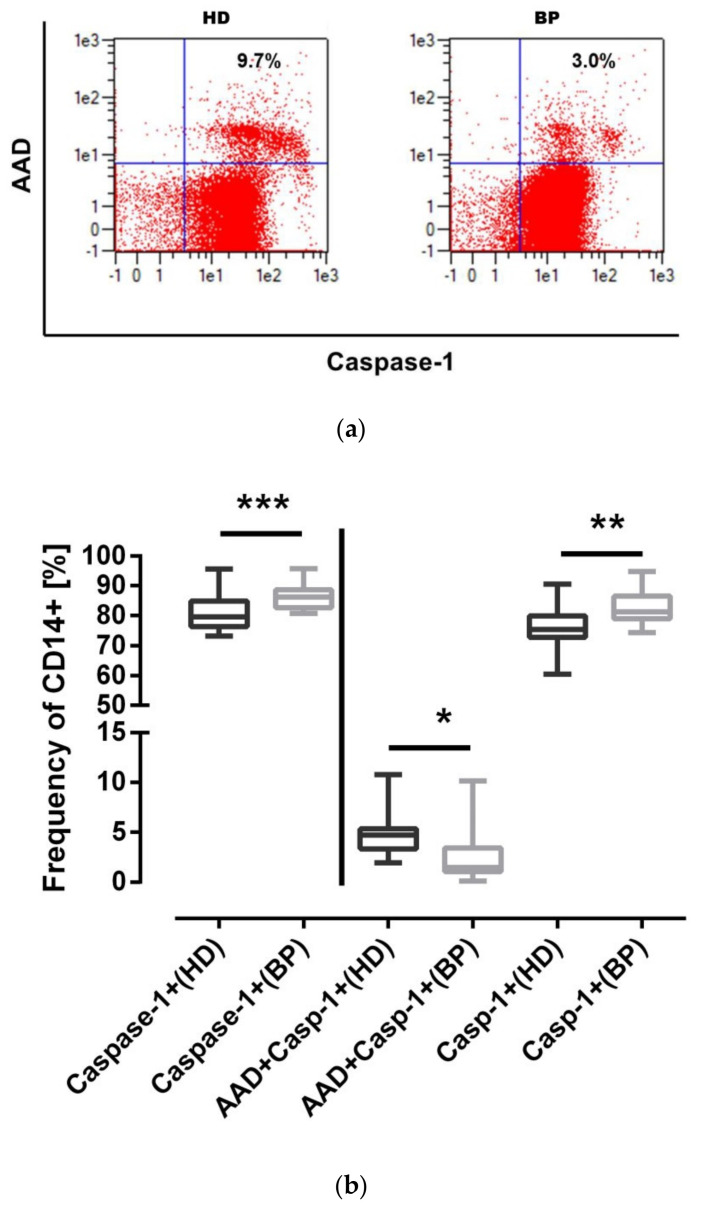
Pyroptosis under basal conditions. (**a**) Representative dot blots depicting pyroptotic cells in the upper right quadrant and CD14-/caspase-1-positive cells in the lower right quadrant; (**b**) on the left hand side is total caspase-1-positivity, and on the right hand side, separated by a vertical line, the more differentiated analysis is depicted showing pyroptotic (CD14+AAD+Casp-1+) and AAD-negative but caspase-1-positive monocytes (Casp-1+) of hemodialysis (HD) patients and patients with high blood pressure (BP). The results are presented as box blots comprising median, 25th, and 75th percentiles. Statistical differences were analyzed by one-way ANOVA, using Friedman analysis as post-test. * *p* < 0.05, ** *p* < 0.01, *** *p* < 0.001.

**Figure 3 toxins-12-00675-f003:**
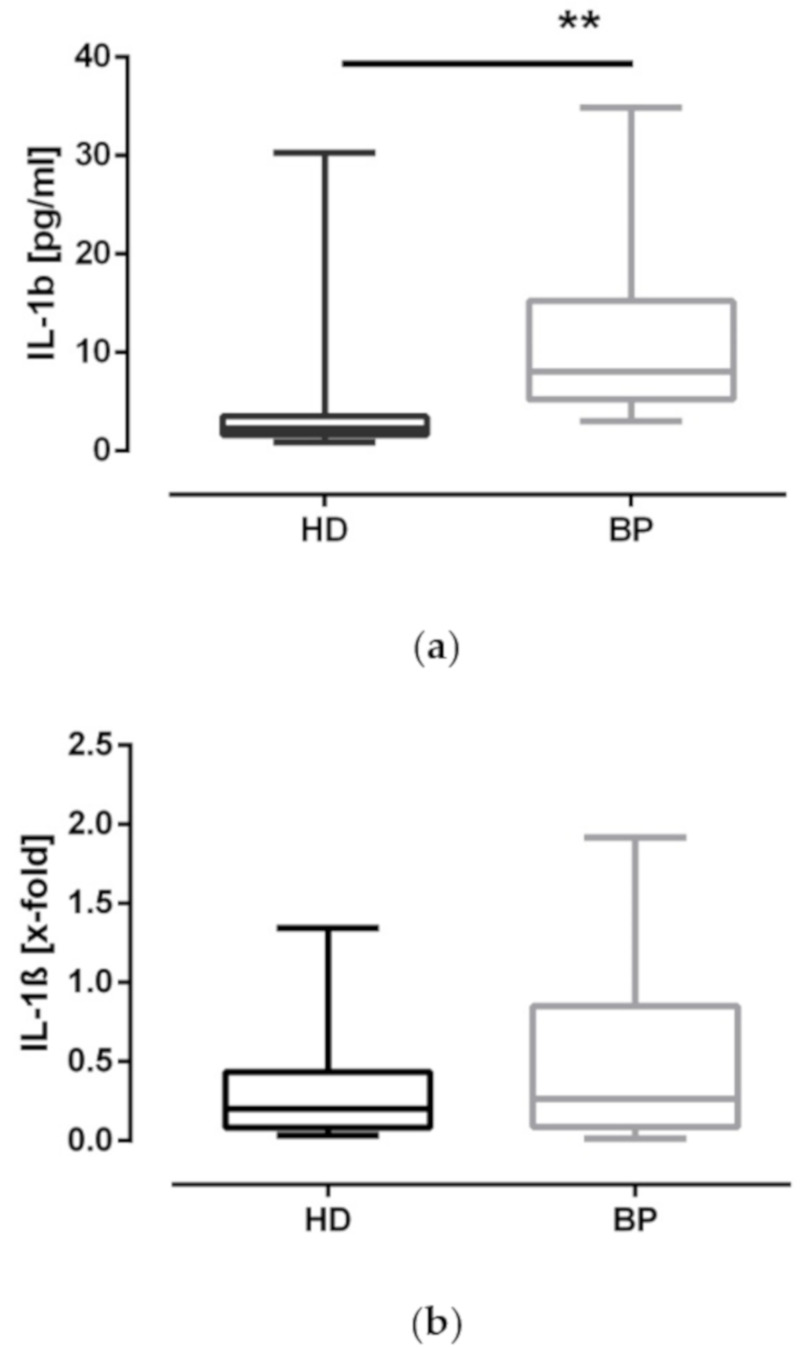
IL-1β protein and mRNA expression. (**a**) IL-1β secretion was determined in the supernatants of cultured PBMCs of hemodialysis (HD) patients and patients with high blood pressure (BP); (**b**) mRNA expression (RQ, x-fold) in PBMCs of HD and BP patients. The results are presented as box blots comprising median, 25th, and 75th percentiles. Statistical differences were analyzed by Wilcoxon test. ** *p* < 0.05.

**Figure 4 toxins-12-00675-f004:**
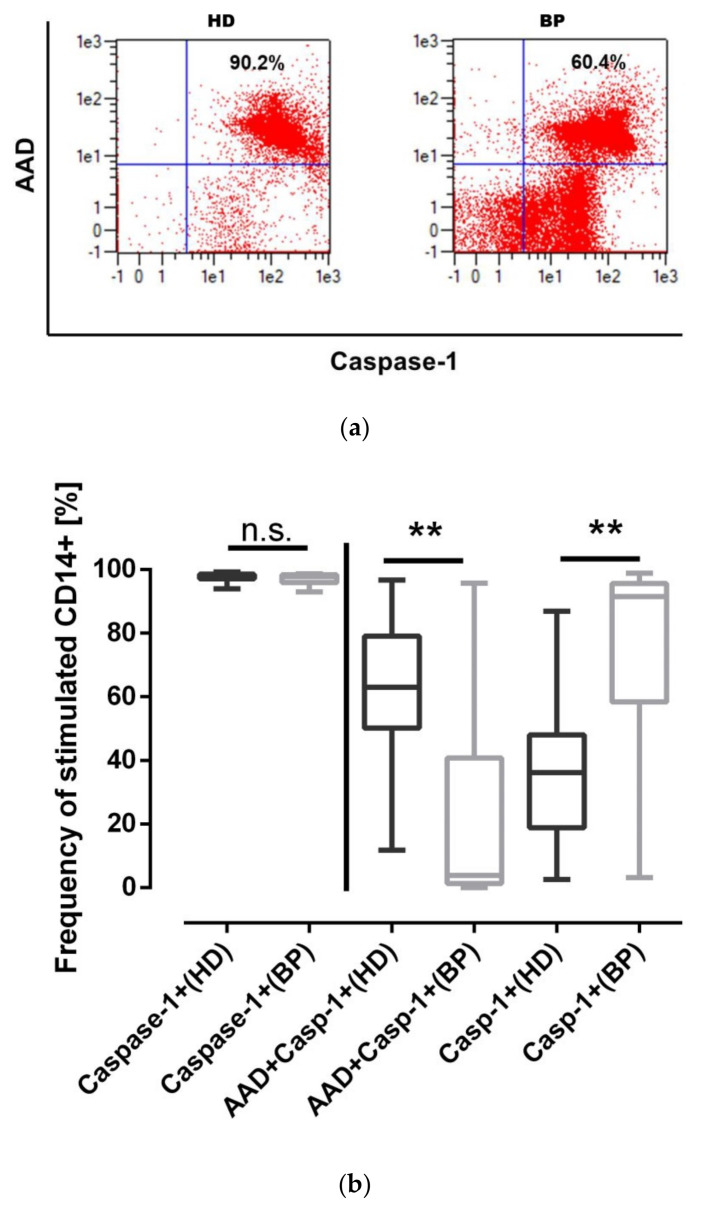
Pyroptosis under stimulatory conditions. (**a**) Representative dot blots depicting pyroptotic cells in the upper right quadrant and CD14-/caspase-1- positive cells in the lower right quadrant; (**b**) on the left hand side is total caspase-1-positivity, on the right hand side, separated by a vertical line, the more differentiated analysis is depicted showing pyroptotic (CD14+AAD+Casp-1+) and AAD-negative but caspase-1-positive monocytes (Casp-1+) of hemodialysis (HD) patients and patients with high blood pressure (BP). The results are presented as box blots comprising median, 25th, and 75th percentiles. Statistical differences were analyzed by one-way ANOVA, using Friedman analysis as post-test, ** *p* < 0.01, n.s.: not significant.

**Table 1 toxins-12-00675-t001:** Patient characteristics.

	HD (*N* = 20)	BP (*N* = 20)	Statistics
Age (years)	57.7 ± 14.3	58.0 ± 12.2	0.864
Gender (female) (%) (*n*)	40 (8)	40 (8)	1.000
BMI (kg/m^2^)	25.9 ± 5.1	29.3 ± 5.7	0.022
Diabetes mellitus (%) (*n*)	10 (2)	10 (2)	1.000
Hypertension (%) (*n*)	100 (20)	100 (20)	1.000
BP (sys., mm Hg)	144 ± 23	147 ± 14	0.880
BP (diast., mm Hg)	84 ± 17	85 ± 5	0.697
Apoplex (%) (*n*)	0 (0)	5 (1)	1.000
CHD (%) (*n*)	10 (2)	10 (2)	1.000
pAD (%) (*n*)	0 (0)	0 (0)	1.000
MI (%) (*n*)	0 (0)	10 (2)	0.487
Smoker (%, ever) (*n*)	60 (12)	35 (7)	0.113
CRP (mg/L)	5.8 ± 11.4	2.0 ± 1.4	0.381
Urea (mmol/L)	24.0 ± 7.4	4.7 ± 1.0	0.001
Creatinine (µmol/l)	926.8 ± 261.6	82.2 ± 12.2	0.001
Hemoglobin (mmol/l)	7.1 ± 0.5	8.8 ± 1.0	0.001
Hematocrit (l/l)	0.3 ± 0.02	0.4 ± 0.04	0.001
Ca (mmol/l)	1.6 ± 0.5	2.4 ± 0.09	0.001
Na (mmol/l)	139.5 ± 3.1	141.4 ± 2.8	0.013
K (mmol/l)	5.2 ± 0.6	4.0 ± 0.5	0.001
Leucocytes (G/µl)	6.6 ± 2.3	6.7 ± 2.1	0.893

Abbreviations: BMI, body mass index; HD, hemodialysis; BP, blood pressure; sys., systolic; diast., diastolic; CHD, coronary heart disease; pAD, peripheral artery disease; MI, myocardial infarction; CRP, C-reactive protein. Continuous variables are expressed as mean ± SD; categorical variables are expressed as percentage. The differences in the two groups were analyzed by paired *t*-test, Wilcoxon test, or Fisher’s exact test.

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
