# Peer review of "NLRP3 Inflammasome Activation in Hemodialysis and Hypertensive Patients with Intact Kidney Function"

_toxins, 2020, doi:10.3390/toxins12110675_

Round 1
Reviewer 1 Report
The authors study the NLRP3 activation in monocytes of 20 HD patients (no renal function) and in 20 hypertensive patients with normal renal function (named BP patients) matched by age, gender and presence of diabetes. In some experiments, they have also included samples of 3 healthy individuals. Via different techniques to estimate the NLRP3 activation(First, Caspase-1 detection, ASC aggregates, pyroptosis degree, IL1B and IL6 expression in resting cells. Second,induction of the activation of the NLRP3 by LPS/nigericin and subsequent analysis of caspase1, IL1B and pyroptosis) they conclude that: (1)In resting state, the monocytes of the BP patients have increased NLRP3 when compared to HD patients.(2) When activated with LPS/nigericin, despite having less basal NLRP3, the monocytes of the HD patients show a higher degree of pyroptosis than those of the BP patients. This suggests that the BP patients have the monocyte NLRP3 activationincreased (compared to HD patients) but the ability of the cells to answer to specific stimuli to promote pyroptosis is decreased.
A role for NLRP3 activation as a link between inflammation and kidney damage has been largely described and some authors suggest that it could be a therapeutic target to prevent CKD. Studies focusing on a better knowledge of NLRP3 activation state in different clinical settings are therefore relevant. The work is well written and the results obtained are interesting. I have some comments for the authors.
Major comments:
1.The results obtained in the pyroptosis analysis result curious to me. In the basal state, all the analysis point out to the fact that NLRP3 is more activated in the BP group (Figure 1 and 2) but then when the pyroptosis (which is the final effect of the NLRP3 activation) is analyzed it turns out that it is decreased in the BP (Figure 2). When the PBMCs are activated with LPS/nigericin, the HD patients show more pyroptosis (Figure 4), more ILB1 and caspase 1 (Figure 4) and more intracellular ATP. Therefore I wanted to point out some issues:
1.1. From my point of view, it is not so clear in which group the NLRP3 is more activated. On one hand, in basal state it effectively seems that it is more activated in the BP group (except for the pyroptosis) and on the other hand the NLRP3 responds much better in stimulated monocytes of HD patients. In my opinion, this dual effect is quite well explained in the abstract but not in the conclusion section. Please address this. Furthermore, the differences in Caspase-1, ASC aggregates and pyroptosis are significant but tight in the resting state. Is it possible that if more patients were included the differences were not seen in the resting state?
1.2. The two groups of patients selected are very different from a renal perspective: HD patients have no renal function and the BP have normal renal function. Do the authors have data of hypertensive patients with some degree of renal disfunction? This would give inside regarding the contribution of kidney disfunction to the NLRP3 activation.
1.3. Regarding the data of the pyroptosis, from a theoretical point of view it should be increased in both HD and BP patients when compared to healthy controls. But in both basal and induced state, pyroptosis of HD patients is increased and decreased in BP patients (vs controls):
-Basal-->Cont: 3.5%, HD: 6.3%, BP: 0.6%. // - Induced-->cont: 44.7%, HD:61.8%, BP: 25.2%.
How do the authors explain this? Please add it to the discussion section.
1.4. From a kidney physiology point of view, the BP patients have high risk to develop kidney disease ending in HD. Thus in this study, the BP patients could be considered a very initial stage of the HD patients. How could it be explained that patients that show a defect in NLRP3 activation (BP patients when induced with LPS) end up to have a better NLRP3 activation (when in HD)?Please discuss it.
1.5. In the discussion section line 189, the authors state that this results can be explained by differences in BMI more than by differences in kidney function. This cannot be assumed at all as this 2 groups of patients are totally different in terms of renal state hence the contribution of the kidney cannot be discarded. Further, HD presence or absence can also have an important impact. Please discuss this.
Minor comments:
- In Figures 2c and 4b, for clarity it should be better indicated what the vertical line means. It is difficult to understand as it is now.
- In the abstract line 5, the sentence ends up with “...are different in both cohorts”. Here the cohorts are not yet introduced and the sentence is confusing. Please rewrite it.
Author Response
Response to Reviewer 1 Comments
Point 1: The Reviewer wonders about the data regarding pyroptosis. While most of the experiments point to a higher activation degree of NLRP3 inflammasome in hypertensive patients with intact kidney function, pyroptosis rates are significantly elevated in haemodialysis patients.
Response 1: We share this view of the Reviewer and tried to substantiate the findings with respect to pyroptosis in our cohorts. “Besides involvement of caspase-1, caspase-8 and caspase-4 may also contribute to NLRP3 activation and pyroptosis (Zheng and Li, 2020, Orning et al., 2019, Gurung et al., 2014, references 20 - 22). Therefore, we measured caspase-8 and caspase-4 transcripts in our cohorts. Caspase-8 for example, interacts with core components of the NLRP3 inflammasome and is required for caspase-1 activation (Gurung et al., 2014, reference 20). Measuring Caspase-8 transcripts we found no differences, neither at the basal state (x-fold expression, HD: 0.7±0.3 vs. BP: 0.8±0.4, p=0.459), nor after canonical stimulation of the NLRP3 inflammasome (x-fold expression, HD: 0.8±0.4 vs. BP: 1.0±0.6, p=0.176). Caspase-4 is able to induce pyroptosis via gasdermin D cleavage, but, caspase-4 transcripts were not different between both groups, too (x-fold basal Casp-4 expression; HD: 1.3±1.4 vs. BP: 1.4±1.1, p=0.671; stimulated HD: 2.8±2.7 vs BP: 2.7±2.2, 0.670). Thus, caspase-8 and caspase-4 most probably do not contribute to elevated pyroptosis measured in HD. But one should also keep in mind that there are still other mechanisms regulating pyroptosis. Rühl et al. could impressively demonstrate that the ESCRTIII-dependent membrane system (endosomal sorting complex required for transport III) can delay or prevent pyroptosis (Rühl et al., Science 2018, reference 23). Therefore, it is quite feasible to hypothesize that in PBMCs of hypertensive patients the ESCRTIII system could be more active in comparison to that of HD patients. This, however, needs further investigations.
We added the additional results to the Result section (basal state: lines 84 – 86; induced state: 146 – 149) and supplemented the materials used to the Material and method section (lines 367 - 368). Interpretations were added to the Discussion section (lines 217 - 228).
Point 1.1.: The Reviewer states that from his point of view it is not quite clear in which group the NLRP3 inflammasome is more activated, for in the resting state of cells BP patients appear to have an higher NLRP3 activity profile whereas in the stimulated state PBMCs of HD appear to respond better to NLRP3-specific stimuli. Therefore, it should be obligatory to give a more differentiated view, not only in the abstract but also in the conclusion section.
Response 1.1.: We totally agree with the Reviewer that the differences in NLRP3 inflammation activation of both cohorts are tight. Therefore, data should be interpreted very carefully. A clear difference between both groups is seen under stimulatory conditions; the pyroptotic rates in HD are dramatically increased compared to BP. However, in the light of the data of Rühl et al. one could also speculate that the repairing system of the ESCARTIII machinery, regulating the amount of gasdermin pores is better developed or perhaps more functional in BP patients, thus preventing that “gasdermin” is instantaneously lytic (Rühl et al., 2018, reference 23).
With regard to a differentiated interpretation of our data (see all Comments 1 -1.5), we would like to present the following conclusion: The NLRP3 inflammasome is activated in overweight patients with hypertension to a higher extent compared to hypertensive dialysis patients. “Whether low pyroptosis rates in PBMCs of BP patients are an indication for an insufficient shut-down of the inflammatory loop or whether high pyroptosis rates of HD patients are part of an over-shooting inflammatory response remains to be clarified.”(Lines 288 – 290).
Abstract section: “Whether low pyroptosis rates are equivalent to a dysfunctional immune response or whether a high pyroptotic output corresponds to over-activation remains to be clarified.” (Lines 19 – 20).
Point 1.2.: The Reviewer remarks that both cohorts in our study are very different from a renal perspective. To get a deeper insight in the role of kidney dysfunction and NLRP3 activation, he proposes to study hypertensive patients with some degree of renal dysfunction.
Response 1.2.: The Reviewer is right; inclusion of patients with different CKD stages would allow a better interpretation of NLRP3 inflammasome activation. Unfortunately, this small pilot study was not designed for studying different CKD cohorts.
Point 1.3.: The Reviewer points to potential interpretations of a control experiment showing pyroptosis in unstimulated (Co: 3.5%; HD: 6.3%; BP: 0.6%) and stimulated PBMCs (Co: 44.7%; HD: 61.8%; BP: 25.2%) of healthy subjects (Co) in comparison to the age-matched HD and BP patients. From a theoretical point of view it appears reasonable to assume that pyroptosis rates are elevated in BP and HD in comparison to healthy subjects.
Response 1.3: We partially agree with the Reviewer.
“At first sight it seems plausible that one should expect elevated pyroptosis rates in the basal state in PBMCs of haemodialysis and hypertensive patients with intact kidney function. Under stimulatory conditions in vitro, there is the possibility for a dysfunctional, low pyroptotic or an overshooting, high pyroptotic response. Induction of the NLRP3 inflammasome with specific stimuli results in “low” (BP), “medium” (Co) and “high” pyroptosis rates (HD) in our patients. But it is quite obvious that a larger study is necessary to answer this question; the number of age-matched controls included in our study is not sufficient to verify this hypothesis.”
We added this paragraph to the discussion section (lines 234 - 240).
Point 1.4.: The Reviewer speculates that from a kidney physiology point of view the hypertensive patients in our study have a high risk to develop kidney disease which may allow for categorizing them as patients at a very early stage of kidney disease. He suggests discussing the “low” pyroptotic rate under stimulatory conditions with regard to the putative developing kidney disease in BP patients.
Response 1.4.: We thank the Reviewer for his suggestion.
“Further on, hypertension is a leading cause for end stage renal failure (Ku et al., AJKD2019, reference 25). Therefore, it is possible that the group of BP patients included in our study may develop renal disease in the future and perhaps these patients already carry undiagnosed signs of kidney disease at a very early stage. On that condition, low pyroptosis rates in BP patients could also be interpreted as adequate “low NLRP3 inflammasome induction. But, of course, from a theoretical point of view, one side of the medal is that a high pyroptosis rate guarantees for total elimination of virus- and / or bacteria-infected cells. The other side of the medal, however, is a kind of over-activated pyroptosis which can result in a massive inflammatory response. Thus, it is not clear if the “lower pyroptosis rate” is good or bad. In point of view of the very small control cohort, the low pyroptosis rate in BP may be too low, but of course the correct interpretation of these results deserves further clarification.”
We added this paragraph to the discussion section (lines 241 - 250).
Point 1.5.: The Reviewer indicates that our data do not allow the conclusion that higher frequencies of CD16+Caspase-1+ monocytes in BP compared to HD are the consequences of an increased BMI in BP patients.
Response 1.5.: We are sorry for the misleading phrasing “that overweight seems to outweigh kidney disease in NLRP3 activation.” Of course, the observation that overweight hypertensive patients have higher frequencies of CD16+Caspase-1+ does not prove causality. For sure, chronic kidney diseases – inclusive haemodialysis treatment – have an impact on NLRP3 activation. The reason for higher frequencies of CD16+Caspase-1+ in BP remains unknown.
Therefore, we substituted the sentence: “Regarding its consequences this observation may be interpreted as an indication that overweight seems to outweigh kidney disease in NLRP3 activation” by:
“The putative mechanism by which hypertension in combination with overweight drives caspase-1 expression in CD16+ monocytes has to be illuminated.” (lines 207 - 209).

Reviewer 2 Report
Overall, the paper presents interesting findings regarding NLRP3 inflammasome activation status in patients that are HD vs BP, but the authors fail to convince the reader of the significance of these results.
Major Flaws:
The groups are described in detail and are matched based on a number of factors but it becomes apparent that obesity may be a contributing factor to these results regardless of blood pressure, yet there is no control to definitively point out what role obesity plays. Normal health controls are also missing for most experiments but for example, when looking at the NLRP3 inflammasome activation they recruit 3 “healthy” patients to assess and compare to the two groups. Why is this the only experiment where healthy individuals were analyzed, could their samples not be used for the other analysis types? Lastly the authors state that the HD group were more likely to be on numerous hypertensive medicines (line 174) yet draw the conclusion that “nevertheless, monoctic NLPR3 activation was even higher in BP…”. This brings up another major flaw as many hypertensive medicines have been shown as having effects on the immune system ( For example Beta Blockers doi: 10.1111/j.1755-5922.2009.00089.x.). To improve upon this study authors need to supply more information on many of these results in comparison to healthy individuals and if possible compare to non-hypertensive obese patients.
Minor Flaws:
Figure 1 does not have a representative dot blot like figures 2 and 4, please add this.
In the supplement please provide full western blots as opposed to the simple western blots provided.
Author Response
Response to Reviewer 2 Comments
Point 1: The Reviewer indicates that there is no adequate control to evaluate the impact of overweight / obesity
Response 1: The study design was chosen to give insight in the putative different NLRP3 inflammasome activation profiles, on the one hand driven by chronic kidney disease and on the other hand by hypertension. The observational study provided an indication that overweight / obesity may be responsible for the higher activation profile in hypertensive patients with intact kidney function in comparison to haemodialysis patients. The Reviewer is right in saying that to gain insight in the mechanisms it is obligatory to include overweight / obese people, and this, of course, will be addressed in a new study.
Point 2: The Reviewer asks why the 3 “healthy subject” were only recruited to assess Inflammasome activation and not for all other experiments
Response 2: The main result of our study is the different pyroptotic output in the basal and especially in the induced state in our two patient groups. To find new clues, we decided to have a look on healthy controls. These data provided the indication that pyroptosis rates in BP are probably too low. But for objective interpretation of the meaning of “low” and “high pyroptosis” more controls have to be included. Of course, one can argue that inclusion of a matched healthy control cohort would have been of advantage, but unfortunately the study design, originally chosen, included only two “arms”. In the revised version of our paper we therefore firstly tried to improve data by measuring caspase-4 and caspase-8 transcripts as both genes are involved one the one hand in NLRP3 inflammasome activation on the other hand in pyroptosis. Both cysteine proteases, however, appear not part of the underlying mechanism (see text 1 below). Secondly, we supplemented the Discussion regarding the observed effects, thus demonstrating that activation of the NLRP3 inflammasome may be different in HD and BP patients as there are specified mechanisms such as the ESCRTIII machinery (endosomal sorting complex required for transport III), which are able to repair Gasdermin-formed pores, thus preventing “over-activated pyroptosis.” The study of ESCRTIII will be part of a new study concept (see text 2 below). Thirdly the conclusion was revised by stating that it is not clear if low pyroptosis rated are equivalent to an immune defect and high pyroptotic power with an over-activated immune response.
1) “Besides involvement of caspase-1, caspase-8 and caspase-4 may also contribute to NLRP3 activation and pyroptosis (references 20 - 22). Therefore, we measured caspase-8 and caspase-4 transcripts in our cohorts. Caspase-8 for example, interacts with core components of the NLRP3 inflammasome and was required for caspase-1 activation (reference 21). Measuring Caspase-8 transcripts we found no differences, neither at the basal state (x-fold expression, HD: 0.7±0.3 vs. BP: 0.8±0.4, p=0.459), nor after canonical stimulation of the NLRP3 inflammasome (x-fold expression, HD: 0.8±0.4 vs. BP: 1.0±0.6, p=0.176). Caspase-4 is able to induce pyroptosis via gasdermin D cleavage, but, caspase-4 transcripts were not different between both groups, too (x-fold basal Casp-4 expression; HD: 1.3±1.4 vs. BP: 1.4±1.1, p=0.671; stimulated HD: 2.8±2.7 vs BP: 2.7±2.2, 0.670). Thus, caspase-8 and caspase-4 most probably do not contribute to elevated pyroptosis measured in HD. (Discussion section: lines 217 – 223; Result section: basal state: lines 84 – 86; induced state: lines 146 - 149).
2) But one should also keep in mind that there are still other mechanisms regulating pyroptosis. Rühl et al. could impressively demonstrate that the ESCRTIII-dependent membrane system (endosomal sorting complex required for transport III) can delay or prevent pyroptosis (Rühl et al., Science 2018, reference 23). Therefore, it is quite feasible to hypothesize that in PBMCs of hypertensive patients the ESCRTIII system could be more active in comparison to that of HD patients. This, however, needs further investigations.” (Discussion section: lines 223 – 228).
3) Whether low pyroptotic rates are equivalent to a dysfunctional immune response or whether a high pyroptotic output corresponds to over-activation remains to be clarified. (Lines 19 - 20).
Point 3: The Reviewer states that hypertensive drug regimen which is significantly different in our groups may have a significant impact on NLRP3 activation.
Response 3: The Reviewer is right. We are sorry not having taken into account the putative effects exerted by beta-blockers. We integrated the findings of Shaw et al. (reference 14) and
replaced the sentence: “Nevertheless, monocytic NLRP3 activation was even higher in BP – the mRNA of IL-1ß which is not expressed in resting myeloid cells of healthy individuals was elevated, higher ASC-speck-formation and higher caspase-1 activity were measured in BP compared to HD patients. Therefore, other reasons than hypertension must be blamed for higher innate immune system activation in BP” by “This, of course, can have an impact on immuno-active cells. As demonstrated by Shaw and colleagues beta-blockers modulate the lymphocytic immune response by down-regulating the expression of CD107 and HLA-DR [14]. The impact of those drugs on monocytes, i.e. NLRP3 inflammasome activation, is unknown and we do not know to what extent the higher NLRP3 activation in BP results from the different drug regime in both groups.” (Lines 188 - 193).
Point 4: The Reviewer remarks that in Figure 1 a representative blot should be presented.
Response 4: We thank the Reviewer of his suggestion and added two dot blots of representing the frequencies of CD14/Caspase-1-positive monocytes in HD and BP (line 92).
Point 5: The Reviewer is of the opinion that instead of Capillary electrophoretic blots (Simple Western) common western blots should be presented.
Response 5: We believe that this is a misunderstanding. Simple western technique is an approved technique (see Qian et al. FASEB J., 2020, doi: 10.1096/fj.201902937RR) alternative to common western blots. Simple Western is a method combining capillary electrophoresis with direct immunostaining. Therefore, both techniques should provide the same results.

Round 2
Reviewer 1 Report
The authors correctly adress most of the issues by adding some new results and better discussing some controversial points.
A minor change from this version:
- "End stage renal failure" should be changed for "End stage renal disease" -- Lines 241-250 discussion section
Further, the minor changes I suggested in the last version are not adressed and I think the authors should do for clarity:
- In Figures 2c and 4b, for clarity it should be better indicated what the vertical line means. It is difficult to understand as it is now.
- In the abstract line 5, the sentence ends up with “...are different in both cohorts”. Here the cohorts are not yet introduced and the sentence is confusing. Please rewrite it.
Author Response
Point 1: The Reviewer suggests replacing the term “End stage renal failure” by the term “End stage renal disease”
Response 1: We agree with the Reviewer and replaced the term accordingly. (Line 51, line 234).
Point 2.: The Reviewer remarks that the meaning of the “vertical line” in Figures 2b and 4b should be explained.
Response 2: We are sorry for the confusing graphical presentation. The vertical line separates the overall caspase positivity (left hand side) and the more differentiated analysis including AAD+Casp-1+ double-positive monocytes and AAD negative but Caspase 2 positive cells (Casp-2) (left hand side). Therefore we included the following statement in the figure legends of figure 2 and figure 4.
…..at the right hand side, separated by a vertical line, the more differentiated analysis is depicted showing pyroptotic (CD14+AAD+Casp-1+) and AAD negative but caspase-1-positive monocytes (Casp-1+) of hemodialysis patients (HD) and patients with high blood pressure (BP). (figure 2: lines 111 - 113; figure 4: lines 163 - 165)
Point 3.: The Reviewer indicates that in the abstract the definition of “both cohorts” should be given.
Response 3: We are sorry for this mistake and changed the sentence: “… in HD and hypertensive patients with intact kidney function (BP).” (Lines 8 - 9).
